# Neural networks for self-adjusting mutation rate estimation when the recombination rate is unknown

Klara Elisabeth Burger[1], Peter Pfaffelhuber[2], Franz Baumdicker[1,3]*

1 Cluster of Excellence "Machine Learning: New Perspectives for Science", University of Tübingen, Tübingen, Germany, 2 Department of Mathematical Stochastics, University of Freiburg, Freiburg, Germany, 3 Cluster of Excellence "Controlling Microbes to Fight Infections", Mathematical and Computational Population Genetics, University of Tübingen, Tübingen, Germany

* franz.baumdicker@uni-tuebingen.de

## Abstract

Estimating the mutation rate, or equivalently effective population size, is a common task in population genetics. If recombination is low or high, optimal linear estimation methods are known and well understood. For intermediate recombination rates, the calculation of optimal estimators is more challenging. As an alternative to model-based estimation, neural networks and other machine learning tools could help to develop good estimators in these involved scenarios. However, if no benchmark is available it is difficult to assess how well suited these tools are for different applications in population genetics.

Here we investigate feedforward neural networks for the estimation of the mutation rate based on the site frequency spectrum and compare their performance with model-based estimators. For this we use the model-based estimators introduced by Fu, Futschik et al., and Watterson that minimize the variance or mean squared error for no and free recombination. We find that neural networks reproduce these estimators if provided with the appropriate features and training sets. Remarkably, using the model-based estimators to adjust the weights of the training data, only one hidden layer is necessary to obtain a single estimator that performs almost as well as model-based estimators for low and high recombination rates, and at the same time provides a superior estimation method for intermediate recombination rates. We apply the method to simulated data based on the human chromosome 2 recombination map, highlighting its robustness in a realistic setting where local recombination rates vary and/or are unknown.

**Data Availability Statement:** All results in this manuscript are reproducible using code available at https://github.com/fbaumdicker/ML_in_pop_gen. This includes the simulation of training data, as well as the adaptive training procedure for the

## Author summary

- single-layer feedforward neural networks learn the established model-based linear estimators for high and low recombination rates

neural network estimators. Furthermore, the optimal coefficients can be computed with a python script available in the same repository.

**Funding:** KB and FB are funded by the Deutsche Forschungsgemeinschaft (DFG, German Research Foundation) under Germany's Excellence Strategy EXC 2064/1 Project number 390727645, and EXC 2124 Project number 390838134. PP is supported in part by the Freiburg Center for Data Analysis and Modeling. We acknowledge support by Open Access Publishing Fund of University of Tübingen. The funders had no role in study design, data collection and analysis, decision to publish, or preparation of the manuscript.

**Competing interests:** The authors have declared that no competing interests exist.

- neural networks learn good estimators for intermediate recombination rates where computation of model-based optimal estimators is hardly possible

- a single neural network estimator can automatically adapt to a variable recombination rate and performs close to optimal

- this is advantageous when recombination rates vary along the chromosome according to a recombination map

- using the known estimators as a benchmark to adapt the training error function improves the estimates of the neural networks

## Introduction

The development of machine learning methods for population genetics faces multiple specific challenges [1]. Nonetheless, it is meanwhile clear that machine learning is a promising technique to build more powerful inference tools. Especially for problems that are hard to tackle down with classical methods they might offer a new approach. In contrast, in population genetics, many theoretical results have been obtained within the last decades and enabled us to identify the best inference technique for specific scenarios. For example, the variance of estimators of the mutation rate, or equivalently the effective population size, is well understood, at least if the rate is constant and recombination is low or high.

### Estimating the effective population size or mutation rate

Estimating the scaled mutation rate, usually denoted by $\theta = 4N_e\mu$, is a fundamental task in population genetics. If the per generation mutation rate $\mu$ is known, estimating the scaled mutation rate corresponds to estimating the effective population size $N_e$, which is often of primary interest and correlates with the genetic diversity in a given population [2]. More precisely, in populations with a small effective population size, genetic drift, i.e. frequency changes due to random sampling, is stronger. Knowledge about $N_e$ allows thus to assess the relative importance of selection, mutation, migration, and other evolutionary forces compared to the influence of genetic drift. The effective population size is often significantly lower than the census population size and varies among populations [3], but also in time [4] and along the genome [5]. The same variation is also observed for the actual mutation rate $\mu$ [6–8]. Consequently, estimating $\theta$, i.e. the mutation rate or $N_e$, is of great interest in many evolutionary fields including conservation genetics, breeding, and population demographics [9].

Many estimators of $\theta$ are developed within the coalescent framework without recombination, where mutations arise along a genealogy given by Kingman's coalescent [10]. However, if recombination is included into the framework [11], estimators of $\theta$ often require an estimate of the recombination rate [12, 13]. One of the most common estimators is Watterson's estimator [14]. It is an easy to compute, unbiased and asymptotically consistent estimator, which has a low variance for high recombination rates, when compared to alternative model-based, unbiased estimators. For Watterson's estimator the only necessary input is the total number of segregating sites in the sample. However, for low recombination rates Watterson's estimator, while still unbiased and consistent, usually has a high variance, when compared to alternative model-based, unbiased estimators. Multiple estimators have been developed based on the site frequency spectrum (SFS), which is the number of segregating sites that occur in $k$ out of $n$ individuals of the sample for $k = 1, \ldots, n − 1$. In particular, without recombination, and if $\theta$ is

known, the coefficients of an optimal unbiased linear estimator based on the SFS have been computed by Fu [15] and the linear estimator that minimizes the mean squared error (MSE) has been identified by Futschik et. al. [16]. Knowledge of the model-based estimators most suitable without and with high recombination rates enables us here to assess and improve the overall performance of artificial neural network estimators which have been trained either for exactly these scenarios or with unknown intermediate recombination rates.

## Machine learning in population genetics

Artificial neural networks are popular in various scientific fields and often used in cases where theoretical models are very complex or hard to analyze [17–19]. Interestingly, machine learning approaches in population genetics, including support vector machines, neural networks, random forests, and approximate Bayesian computation (ABC) often use summary statistics of the genetic data borrowed from the theoretical literature as input data [1, 20], such as the site frequencies. However, there is an increasing number of studies and methods that do not rely on summary statistics. One example is a recent publication by Flagel et al. [21] who showed that effective population genetic inference can also be reached with deep learning structures like convolutional neural networks (CNNs) without precomputed summary statistics. Instead of a set of summary statistics they used the full genotype matrix as input for their CNN. The CNNs performed surprisingly good in different tasks from population genetics, including inferring historic population size changes and selection pressures along the genome. The genotype matrix has also been used as input data in other studies using deep learning in population genetics [22, 23]. As, in contrast to image data, the rows of the genotype matrix can be shuffled without loosing information, an adaptation of the network architecture or presorting the genotype matrix is often beneficial. One approach is suggested in [24] introducing an exchangeable neural network for population genetic data that makes the ordering of the samples invisible to the neural network. Besides summary statistics and the genotype matrix, tools based on inferred gene trees and ancestral recombination graphs are emerging [25]. Recently, Sanchez et al. [23] showed that combining deep learning structures using the genotype matrix and approximate Bayesian computation provides an effective method to reconstruct the effective population size through time for unknown recombination rates. Thus, artificial neural networks are a promising tool in more involved scenarios where theoretical insights are harder to obtain, although the often complex architectures impair the comprehension of the underlying learning process.

Here, we take a different perspective and consider the estimation of the mutation rate from single nucleotide frequencies. If data is generated within a coalescent framework, the optimal estimator (which is a linear map of the SFS) is known for no recombination when $\theta$ is known, and for high recombination rates. In particular, we consider a dense feedforward neural network with at most one hidden layer, which already suffices to achieve almost the performance of the optimal linear estimators and at the same time to provide a superior estimator for variable recombination rates.

## Materials and methods

### Model-based estimators of mutation rate and population size

In this section, we recall the properties of known estimators for $\theta$, that are linear in the site frequency spectrum. More precisely, we consider mutations as modeled by a neutral Wright Fisher model with infinitely many sites model along Kingman's coalescent. The model-based estimators presented below were proposed and analyzed by Watterson [14], Fu [15] and

Futschik et al. [16]. In order to give a self-contained presentation, we now recall some basics on the coalescent and give details of the above estimators:

To obtain Kingman's coalescent for a sample of size $n$ we trace back the series of ancestors through time. Therefore, we define an ancestral tree for this sample by repeatedly coalescing each pair of two lineages at a rate 1, such that, when $k$ is the number of remaining lineages, at rate $\binom{k}{2}$ two randomly chosen lineages coalesce. The resulting random tree is called Kingman's coalescent. We denote the random duration the coalescent spends with exactly $k$ lineages by $T_k \sim \text{Exp}\left(\binom{k}{2}\right)$. Neutral mutations are independently added upon this tree.

More precisely, for a given genealogical tree, mutations can happen everywhere along the branches at rate $\frac{\theta}{2}$, see Fig M in S1 Text. Given the length $\ell$ of a branch, the number of mutations on this branch is $\text{Poi}\left(\frac{\theta}{2}\ell\right)$ distributed. Consequently there are $M \sim \text{Poi}\left(\frac{\theta L}{2}\right)$ mutations along the tree, if $L = \sum_{k=2}^{n} kT_k$ is the total length of the tree. We consider the infinitely many sites model, where each mutation hits a new site. Thus the frequency of the derived allele in the sample population is given by the number of descendants of the branch where the mutation occurred. We define the site frequency spectrum as $S = (S_1, \ldots, S_{n-1})$, where $S_i$ is the number of mutations that occur on a branch that is ancestral to exactly $i$ out of $n$ individuals in the sample.

In the following, we are looking for estimators of the form

$$\hat{\theta} = \sum_{i=1}^{n-1} a_i S_i.$$

which are uniquely defined by the vector $a = (a_1, \ldots, a_{n-1})$. We will see that optimal choices for $a$ frequently depend on $\theta$, which will lead to an iterative estimation procedure.

**Unbiased linear estimators of the mutation rate $\theta$.** Watterson's estimator is given by setting $a_1 = \cdots = a_{n-1} = \mathbb{E}[L]^{-1}$, i.e.

$$\hat{\theta}_{\text{W}} := \frac{M}{h_n} = \sum_{i=1}^{n-1} \frac{S_i}{h_n} \quad \text{with} \quad h_n := \sum_{i=1}^{n-1} \frac{1}{i}$$

being the $n$-th harmonic number and $S = (S_1, \ldots, S_{n-1})$ the site frequency spectrum.

From a theoretical point of view, only the cases of no recombination or in the limit of high recombination (leading to independence between loci) can easily be treated. Watterson's estimator is unbiased for $\theta$ for all recombination rates and if there is no recombination the variance, as found by Watterson [14], is given by

$$\mathbb{V}_{\theta}\left[\hat{\theta}_{\text{W}}\right] = \theta \frac{1}{h_n} + \theta^2 \frac{g_n}{h_n^2} \quad \text{where} \quad g_n := \sum_{i=1}^{n-1} \frac{1}{i^2}.$$

In the limit of high recombination (unlinked loci) we get that $S_k \sim \text{Poi}\left(\frac{\theta}{k}\right)$ and $(S_1, \ldots, S_{n-1})$ are independent, such that the variance reduces to

$$\mathbb{V}_{\theta}\left[\sum_{i=1}^{n-1} \frac{S_i}{h_n}\right] = \frac{\theta}{h_n}.$$

In this case, using standard theory on exponential families, $\sum_{i=1}^{n-1} S_i$ is a complete and sufficient statistic for $\theta$, and it follows from the Lehmann–Scheffé theorem that Watterson's estimator is a unique Uniformly Minimum Variance Unbiased Estimator (Uniformly MVUE) [26]. However, this only holds for unlinked loci.

If no recombination is included the MVUE estimator was found by Fu [15]. In this case, the best linear unbiased estimator of $\theta$ from the site frequency spectrum $S = (S_1, \ldots S_{n-1})$ is given in matrix notation by

$$f_\theta(S) = \frac{\alpha^\top (D_\alpha + \theta\Sigma)^{-1}}{\alpha^\top (D_\alpha + \theta\Sigma)^{-1}\alpha}\, S,$$

whereby

$$\alpha = (\alpha_1, \ldots, \alpha_{n-1}) = \left(1, \frac{1}{2}, \ldots, \frac{1}{n-1}\right), \quad D_\alpha = diag(\alpha_1, \ldots, \alpha_{n-1}) \quad \text{and}$$
$$\Sigma = \{\sigma_{ij}\}, \qquad i, j = 1, \ldots, n-1,$$

symmetric with

$$\sigma_{ii} := \begin{cases} \beta_n(i+1) & \text{if } i < \frac{n}{2}, \\ 2\dfrac{h_n - h_i}{n-i} - \dfrac{1}{i^2} & \text{if } i = \frac{n}{2}, \\ \beta_n(i) - \dfrac{1}{i^2} & \text{if } i > \frac{n}{2}, \end{cases}$$

and for $i > j$

$$\sigma_{ij} := \begin{cases} \dfrac{\beta_n(i+1) - \beta_n(i)}{2} & \text{if } i+j < n, \\ \dfrac{h_n - h_i}{n-i} + \dfrac{h_n - h_j}{n-j} - \dfrac{\beta_n(i) + \beta_n(j+1)}{2} - \dfrac{1}{ij} & \text{if } i+j = n, \\ \dfrac{\beta_n(j) - \beta_n(j+1)}{2} - \dfrac{1}{ij} & \text{if } i+j > n, \end{cases}$$

where

$$\beta_n(i) := \frac{2n(h_{n+1} - h_i)}{(n-i+1)(n-i)} - \frac{2}{n-i}.$$

We note that the optimal coefficients $a(\theta) = (a_1(\theta), \ldots, a_{n-1}(\theta))$ depend on the real $\theta$. In practice, therefore, the estimation of $\theta$ depends on an iterative procedure that approximates Fu's estimator as described below. The iterative version is neither linear nor unbiased, but close to the non-iterative version (Fig H in S1 Text).

**General linear estimators of the mutation rate $\theta$.** So far we only considered unbiased estimators and minimized their variance. Allowing for a potential bias of the estimator can decrease the MSE when compared to the unbiased estimators. The SFS based estimator with minimal MSE in the absence of recombination was found by Futschik and Gach [16]:

The linear estimator of $\theta$ from the site frequency spectrum $S = (S_1, \ldots S_{n-1})$ with minimal MSE is given by

$$\tilde{f}_\theta(S) = \alpha^\top \left(\frac{D_\alpha}{\theta} + \Sigma + \alpha\alpha^\top\right)^{-1} S,$$

where $\alpha$, $D_\alpha$, and $\Sigma$ are as above.

Note that Futschik and Gach introduced multiple variants for the estimation of $\theta$. The estimator $\tilde{f}_\theta(S)$ used here corresponds to formula (26) in Futschik and Gach [16]. Another variant

from Futschik and Gach is a modification of Watterson's estimate that minimizes the MSE for estimates based on the number of segregating sites $\sum_{i=1}^{n-1} S_i$ in the scenario without recombination (Fig O in S1 Text). For positive recombination rates further variants depend on estimates of the recombination rate.

**Iterative estimation of $\theta$.** The estimators of Fu and Futschik depend on the true but unknown $\theta$. In practice, we thus have to build an iterative estimator, which will yield a $\theta$-independent estimator and approximate the variance minimizing or MSE-minimizing estimators. Starting with some $\hat{\theta}_0$, e.g. $\hat{\theta}_0 = \hat{\theta}_W$, Wattersons estimator, we set

$$\hat{\theta}_{k+1}(S) := f_{\hat{\theta}_k}(S)$$

which usually converges quickly. For example, for $n = 40$ and $\theta = 40$ it takes about 5 iterations until the estimate of $\theta$ is obtained up to 3 decimal places.

We call the resulting estimator $\hat{\theta}_{ItV}$ for Fu's estimator $f_\theta$, and $\hat{\theta}_{ItMSE}$ for Futschik's estimator $\tilde{f}_\theta$. Note that due to the iteration the estimators are no longer explicitly linear or unbiased, but do not depend on $\theta$.

## Estimating the mutation rate with a dense feedforward neural network

We trained dense feedforward neural networks with zero or one hidden layer to perform the estimation task. Note that the architecture of the neural networks determine the type of functions the neural network can approximate. For example, if no hidden layer and no bias is included, the architecture ensures that the resulting estimator is a linear function in $S = (S_1, \ldots, S_{n-1})$.

**Simulation of training data.** In contrast to model-based estimators, the estimators first have to be trained in order to optimize parameters within the neural network. Therefore, we rely on simulations to train the neural network and then evaluate the resulting estimators. Five training data sets were generated with the software msprime by Kelleher et al. [27] for various parameters. For each training data set with $2 \cdot 10^5$ independent site frequency spectra, the haploid sample size is given by $n = 40$ and the recombination rate $\rho$ was either set to 0 (no recombination), 20, 35 (moderate recombination), or 1000 (high recombination). Hereby, the recombination rate is scaled by $N_e$, such that $\rho = rL4N_e$, where $r$ is the per generation per site recombination rate and $L$ is the length of the simulated sequence. Within all training data sets the mutation rate $\theta$ is chosen uniformly in $(0, 100)$. In addition, we combined the data sets with no recombination and high recombination with a data set with a uniformly chosen recombination rate in $(0, 50)$. This creates the fifth training data set of total size $6 \cdot 10^5$ with a variable recombination rate.

**Feedforward neural networks.** In this project we consider two artificial neural networks: a linear dense feedforward neural network (no hidden layer) and a dense feedforward neural network with one hidden layer and an adaptive loss function to ensure a robust performance. All neural networks take $S$ as input and are trained for $\theta$ chosen uniformly in $(0, 100)$. ReLU was used as activation function and as optimizer the Adam algorithm [28] was chosen. The neural networks have been implemented in Python 3 via the library tensorflow and keras. Details of the training procedure and hyperparameter choice are given in section B in S1 Text. The code is made available on Github: fbaumdicker/ML_in_pop_gen.

**Linear neural network.** As a simple check whether the network is able to learn the mutation rate at all we considered a neural network with no hidden layer and no bias node included, i.e. a linear neural network. This simple network structure ensures that the estimator is linear in $S$, but not necessarily unbiased.

**Neural network with one hidden layer.**   To investigate how more complexity in the architecture of the neural network improves the performance, we also implemented a neural network with one hidden layer and one bias node. See Fig L in S1 Text for visualization. Naturally, the question arises on how many hidden nodes to include in this additional layer. We observed that the optimal number of nodes depends on the sample size $n$. Performace for larger $n$ improved when using more hidden nodes. In our experience, it is advisable to use at least $2n$ nodes. Using too few hidden nodes introduces an increased risk of losing robustness and using less than $n$ nodes very often results in non-termination of the training process. Using significantly more hidden nodes produced similar results, but the runtime can increase significantly. For $n = 40$, using 200 nodes in the hidden layer has proven to be a good choice.

**Adaptive reweighting of the loss function by model-based estimators.**   All neural networks have been trained on simulated data where $\theta$ is uniformly chosen in $(0, 100)$. From the linear model-based estimators in absence of recombination we know that the coefficients of the optimal estimator depend on $\theta$. Hence, within the neural networks, we included an adaptive reweighting of the training data with respect to the parameter $\theta$. This ensures that the inferred estimator is not worse than the iterative estimators of Fu and Futschik nor Watterson's estimator for all possible values of $\theta$. A visualization of the training procedure is shown in Algorithm 1. The training of the "adaptive" neural network is done in several iterations. Each iteration consists of training a neural network as before and subsequently increasing the weight of those parts of the training data where the normalized MSE (nMSE), i.e. the MSE divided by $\theta$, is not close enough to the minimal nMSE of the iterative versions of Fu's and Futschik's, Watterson's, and the linear neural network estimator. Note, this evaluation in between the training steps requires a second validation data set, in addition to the one used to train the neural networks themselves.

For this comparison we divided the validation and training data into six subsets with respect to $\theta$. The borders of the subsets are defined by $(t_i)_{i=0,\ldots,6}$, where $t_0 = 0$, $t_1 = 1$ and $t_0 < t_1 < \cdots < t_6 = 100$. In the $k$-th subset we let $t_{k-1} < \theta \leq t_k$. The $t_k$ are chosen such that the range of coefficients $a_j(\theta)$ in Fu's estimator is the same in the subsets, i.e.

$$\sum_{j=1}^{n-1} a_j(t_k) - a_j(t_{k-1}) \tag{1}$$

yields the same value for all $1 < k \leq 6$. Fig K in S1 Text illustrates the chosen interval borders. The total loss function is then given by

$$\frac{1}{m} \sum_{i=1}^{m} \left( \frac{\hat{\theta}_i - \theta_i}{\theta_i} \right)^2 \cdot \omega(\theta_i) \tag{2}$$

with $\omega(\theta_i) = \sum_{k=1}^{6} \omega_k \cdot 1_{\{t_{k-1} < \theta_i \leq t_k\}}$. In particular, the loss function can penalize errors in some subsets more than in others.

The weights $\omega_k$ are initialised by 1 and $\omega_k$ is updated in every iteration of comparable poor performance by setting

$$\omega_k = \omega_k + R \cdot \frac{\max\left(\frac{D_k}{b_k}, 0\right)}{\max_k\left(\max\left(\frac{D_k}{b_k}\right), 0\right)} \tag{3}$$

with

$$b_k := \min(\mathrm{nMSE}_k(\hat{\theta}_{\mathrm{W}}), \mathrm{nMSE}_k(\hat{\theta}_{\mathrm{ItV}}), \mathrm{nMSE}_k(\hat{\theta}_{\mathrm{ItMSE}}), \mathrm{nMSE}_k(\hat{\theta}_{\mathrm{LinearNN}})),$$

$$D_k := \mathrm{nMSE}_k(\hat{\theta}_{\mathrm{ANN}}) - b_k,$$

where $\mathrm{nMSE}_k(\hat{\theta})$ is the empirical normalized MSE of $\hat{\theta}$ on the subset where $t_{k-1} < \theta \leq t_k$ and $R$ is drawn uniformly at random between 0.25 and 0.5.

The training is finished as soon as an iteration does not result in a weight update, i.e. the adaptive neural network performs comparable or better than the model-based estimators and the linear neural network on each of the six subsets or to be precise

$$\mathrm{nMSE}_k(\hat{\theta}_{\mathrm{ANN}}) \overset{!}{\leq} 1.02 \cdot b_k \tag{4}$$

for $k = 1, \ldots, 6$. In principle, if the network architecture does not provide enough capacity, i.e. if the number of hidden nodes is too low, the condition in (4) can lead to longer runtimes or prevent a termination. In our scenario, using 200 hidden nodes, this rarely occurred.

**Algorithm 1: Overview on adaptive training procedure.** This pseudocode illustrates the basic principle of the adaptive training procedure used for the adaptive neural networks.

---

**input**  : $n$, classes, data
**output**: trained adaptive NN

**obtain** : six classes for adaptive training via Eq (1)
**split**  : data into training and validation data
**obtain** : nMSE for $\hat{\theta}_{\mathrm{W}}, \hat{\theta}_{\mathrm{ItV}}, \hat{\theta}_{\mathrm{ItMSE}}$ & $\hat{\theta}_{\mathrm{LinearNN}}$ for each class on validation data
**obtain** : $b$ containing smallest nMSE of $\hat{\theta}_{\mathrm{W}}, \hat{\theta}_{\mathrm{ItV}}, \hat{\theta}_{\mathrm{ItMSE}}$ & $\hat{\theta}_{\mathrm{LinearNN}}$ per class
**init**   : loss weights $\omega(\theta) = 1$ and $\mathrm{nMSE}(\hat{\theta}_{\mathrm{NN}}) \gg b$

**while not** $nMSE(\hat{\theta}_{NN}) \leq 1.02 \cdot b$ *in each class (cond. (4))* **do**
 **train**  : NN via adaptive loss function (2)
 **obtain** : $\mathrm{nMSE}(\hat{\theta}_{\mathrm{NN}})$
 **update:** update loss weights $\omega(\theta)$ for each class as shown in Eq (3)

---

## Results

To investigate the capabilities of neural networks, two dense feedforward neural networks, with zero or one hidden layer, as a function of the site frequency spectrum were considered in this work. Those fairly simple network architectures were consciously chosen to facilitate a better understanding of the underlying learning process of the neural networks. Additionally, neural networks as a function of the site frequency spectrum can be easily compared to the model-based estimators, which use the same information. We observed multiple properties in our simulations:

If no recombination is included in the training and test data, as in Fig 1A, the neural nets perform comparably to Futschik and Gach's estimator, which has the lowest MSE among linear estimators in this scenario. If the recombination rate is high in the training and test data, as in Fig 1D the neural networks perform comparably to Watterson's estimator, which is a uniformly MVUE for $\theta$ in this situation. For training and test data sets with moderate recombination (Fig 1B and 1C), the neural networks outperform the other estimators. The bias of the adaptive neural network estimator is higher for smaller values of $\theta$ than for larger $\theta$, but always smaller than the bias of the estimator of Futschik and Gach, see Fig J in S1 Text.

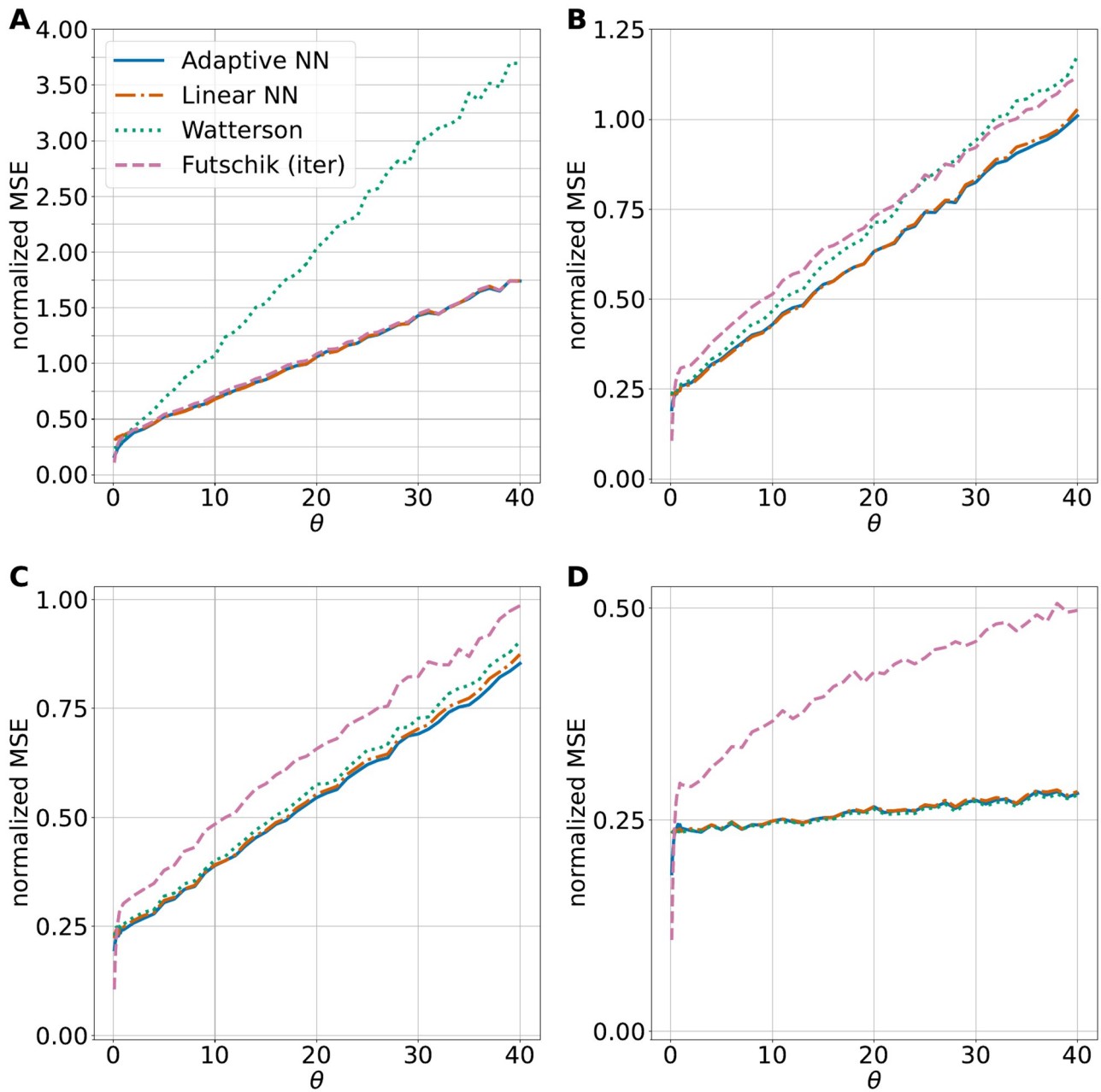

**Fig 1. Performance of mutation rate estimators.** The normalized MSE for four different estimators are shown: The iterated minimal MSE estimator (Futschik (iter)), Watterson's estimator, and two neural network estimators. A linear network without a hidden layer and an adaptive network with one hidden layer and 200 hidden nodes. For each subplot, we used msprime to generate a total of $4.9 \cdot 10^5$ independent simulations with sample size $n = 40$, recombination rate $\rho$, and 49 different mutation rates $\theta \in (0, 40]$. A: recombination rate $\rho = 0$, B: $\rho = 20$, C: $\rho = 35$, D: $\rho = 1000$. All shown neural networks have been trained on data sets with the corresponding recombination rate $\rho$.

So far the considered neural network estimators depend on the recombination rate as the neural networks are trained on data sets with the same recombination rate as the test data. However, the recombination rate is often unknown or varies along the genome sequence such that a method that has a low variance regardless of the local recombination rate is desirable. To see if the neural networks can achieve this we trained them on data with variable

recombination rates. In this case, due to the restriction to linear estimators, the linear neural network is not able to match the performance of the adaptive neural network, especially for low and high recombination rates, but is surprisingly good for intermediate recombination rates. In contrast, the non-linear neural network with one hidden layer produced an estimator that had the lowest MSE for almost all possible values of the recombination rate (see Fig 2). This is for example beneficial when a sliding window approach is used to estimate the local mutation rate along a genome sequence. We illustrate this case applying the estimator to data based on the human recombination map. There the estimate depends on the local recombination rate, such that the neural network trained for variable recombination has the advantage to produce good estimates without an explicit estimate of the recombination map.

Naturally, the question of what has been learned by the neural network arises. To answer this question, one usually tries to draw conclusions from the learned weights, but this is often not trivial. Here, we used SHAP [29] to compute and analyze the feature importances of the linear and adaptive neural network, see section A in S1 Text. The importance of mutations in low frequency, i.e. $S_i$ for small $i$, was larger than the importance of high frequency mutations. For the latter, however, the network has learned to take positive values of $S_i$ with larger $i$ differently into account depending on the recombination rates. A more detailed consideration of the SHAP values is given in section A in S1 Text.

## Comparison with alternative ML methods

In this section, we compare the adaptive neural network to more sophisticated ML methods. For this purpose, we consider the convolutional neural network by Flagel et al. [21] and a standard Approximate Bayesian Computation (ABC) approach. The CNN introduced by Flagel et al. were applied to a number of evolutionary questions including identifying local introgression, estimating the recombination rate, detecting selective sweeps, and inferring population size changes. As a test case, Flagel et al. also used a CNN to infer $\theta$ based on the genotype matrix. We trained this CNN based on the same data sets as the other neural networks considered here. While the genotype matrix carries more information than the site frequency spectrum, e.g. about joint occurrence of mutations, it might be easier to learn some important features directly from summary statistics. ABC is a method based on Bayesian statistics to estimate the posterior distribution of model parameters. The basic idea of ABC is first to approximate the likelihood function using simulations and then to compare the outcome with the observed data. For our comparison we used the rejection method of the R package abc [30] based on the training data sets of the adaptive neural network, i.e. with $2 \cdot 10^5$ simulations to estimate $\theta$.

Fig 3 compares the adaptive neural network to the CNN, the ABC method and Watterson's estimator in the same setting as before. Both, the CNN and ABC were trained for $\theta \in (0, 100)$, as the adaptive neural network, and especially struggle for small $\theta$ values. The CNN stabilizes for larger $\theta$, whereas the ABC based estimator does not. Even though the CNN stabilizes for larger $\theta$, we generally did not observe a stable performance, as Fig N in S1 Text shows. In contrast, the adaptive neural network is more robust due to the adaptive training procedure, a comparison is included in Fig N in S1 Text. In all four subplots of Fig 3 the adaptive neural network represents the preferred choice. This highlights the capabilities of the adaptive comparison with model-based estimators despite the architectural simplicity of the neural network.

## Application

If a larger genome sequence is considered, the recombination rate varies between different regions of the chromosomes. We applied the trained neural networks and model-based

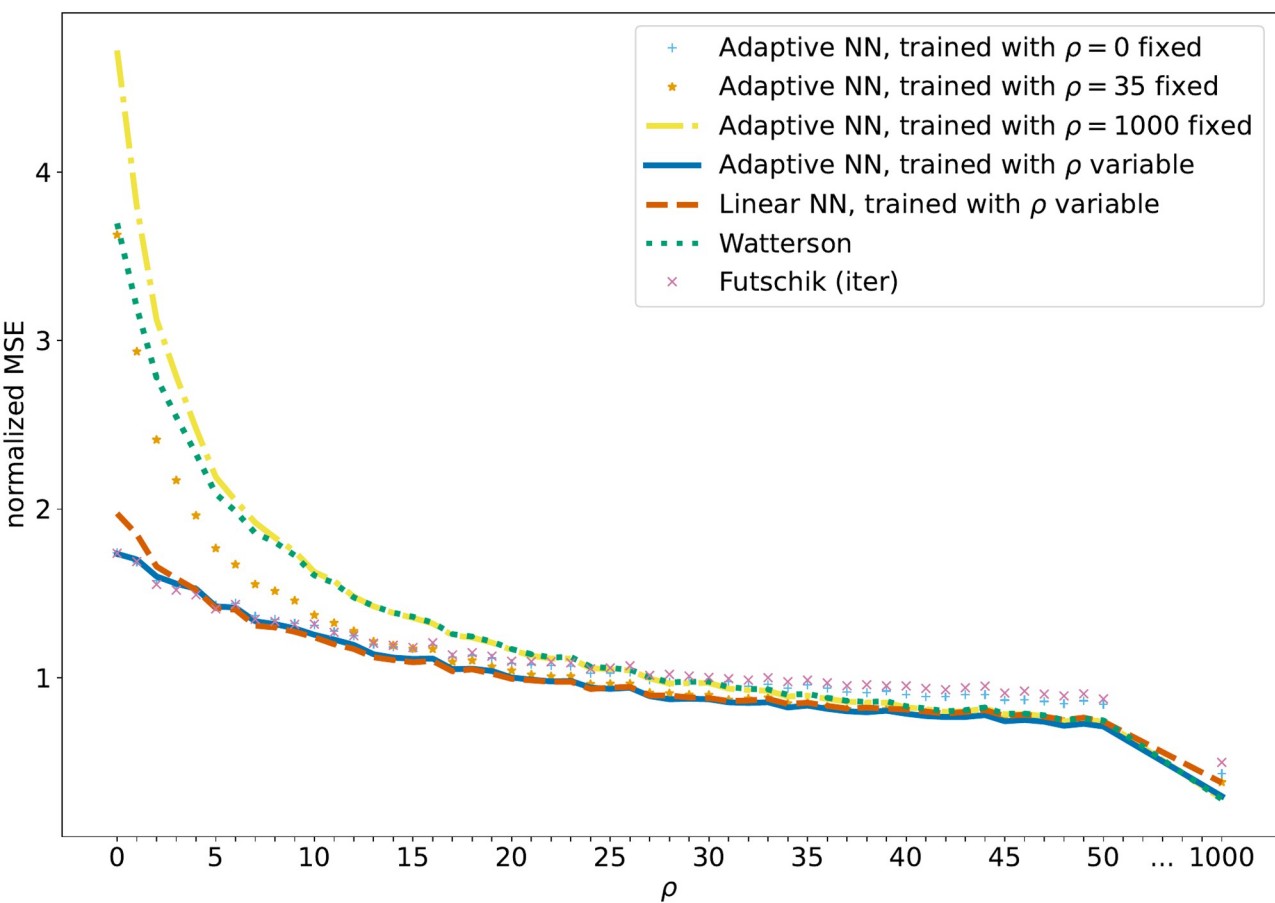

**Fig 2. MSE for variable recombination rates.** The normalized MSE of feedforward neural network estimators of the mutation rate compared to model-based estimators. In particular, the adaptive networks trained with a fixed recombination rate are compared to the networks trained with variable recombination rates. The recombination rate is given by $\rho = 0, \ldots, 50$ or $\rho = 1000$, while the sample size $n = 40$ and mutation rate $\theta = 40$ are fixed. MSE estimates are based on $10^5$ independently simulated SFS for each value of $\rho$.

estimators in this more realistic setting. For this purpose, data for human chromosome 2 (chr2) were generated using stdpopsim [31] based on the HapMap human recombination map [32], see Fig 4. The estimators were applied in a sliding window approach along the chromosome. Using a constant per generation mutation rate of $1.29 \cdot 10^8$ per base pair [33] and a window size of 70kb this resulted in a scaled mutation rate of $\theta = 36.12$ and a scaled local recombination rate between 0 and 1622 illustrated in Fig 4. The estimates of $\theta$ along the chromosome are shown in Fig 5. Close to chromosome position $1.0 \cdot 10^8$ the impact of the dependency on a single tree due to no recombination is observable. Watterson's estimator often resulted in a larger deflection from the true $\theta = 36.12$ than the other estimators. This is in particular true for regions with low or no recombination. If multiple windows fall within one such region the corresponding estimates are strongly correlated. For the region of low recombination near chromosome position $1.0 \cdot 10^8$ Fig P and Q in S1 Text show a zoomed-in version of the recombination map in Fig 4 and the estimates of $\theta$ in Fig 5. To better illustrate this effect, the estimators have been applied to data generated by an artificial recombination map with multiple regions without recombination separated by regions with recombination, see section C in S1 Text.

A separate consideration of disjoint windows with low ($0 < \rho \leq 1$), intermediate ($30 < \rho \leq 40$), or high ($\rho > 150$) recombination according the the human recombination map is shown

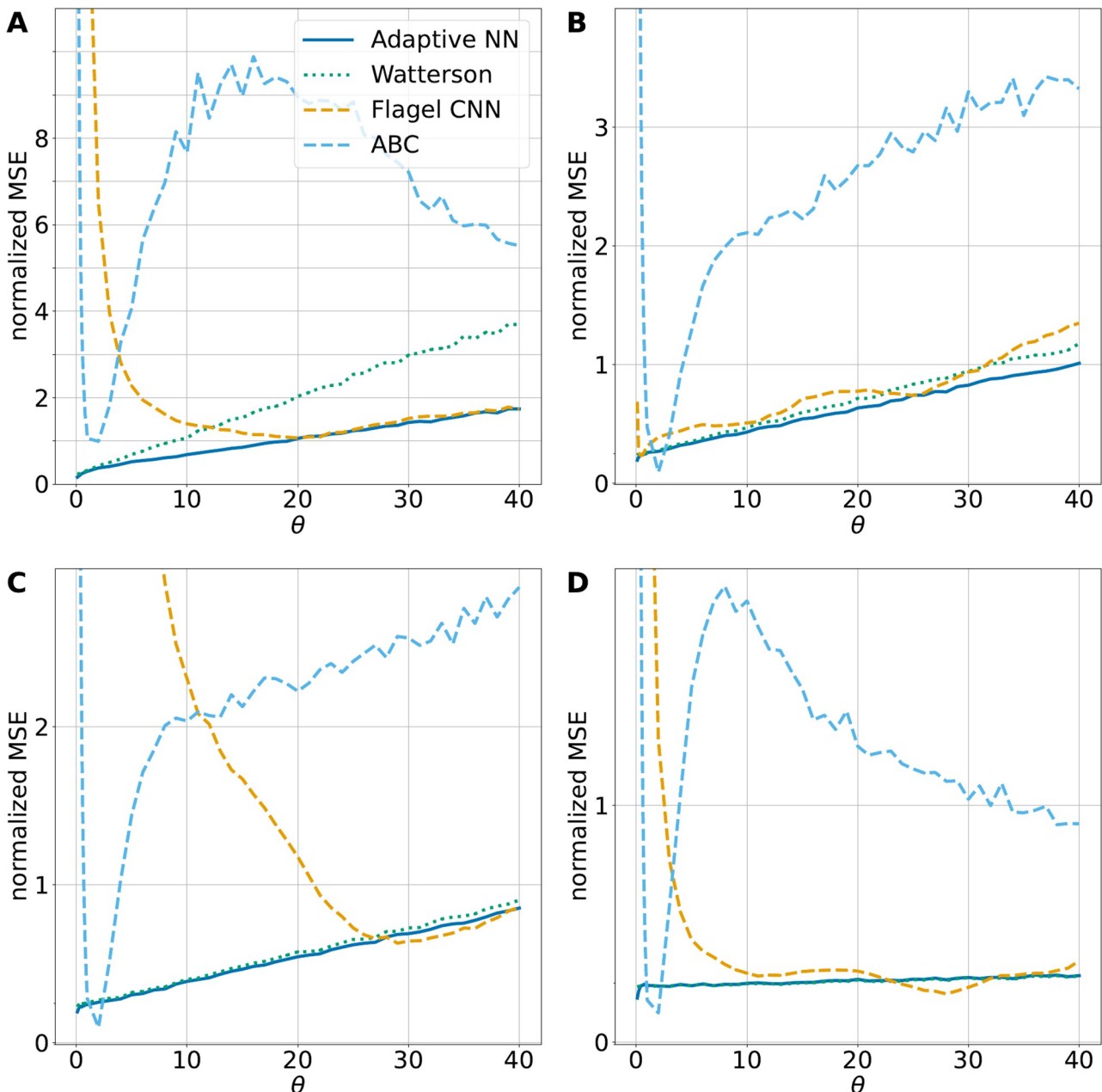

**Fig 3. Comparison to other ML methods.** The normalized MSE for four different estimators are shown: Watterson's estimator and the adaptive neural network (as in Fig 1), a retrained CNN by Flagel et al. [21] and Approximate Bayesian Computation (ABC). The same simulations as in Fig 1 have been used. A: recombination rate $\rho = 0$, B: $\rho = 20$, C: $\rho = 35$, D: $\rho = 1000$. All shown neural networks and ABC have been trained on data sets with the corresponding recombination rate $\rho$.

in Fig 6. The observations in the constant recombination setting are also present in this more realistic scenario. Watterson's estimator more often strongly overestimates theta for low and medium recombination rates while Futschik's estimator performs better for low than high recombination rates. The neural networks trained with variable $\rho$ is also in this more realistic scenario based on a recombination map able to adapt to the local recombination rate.

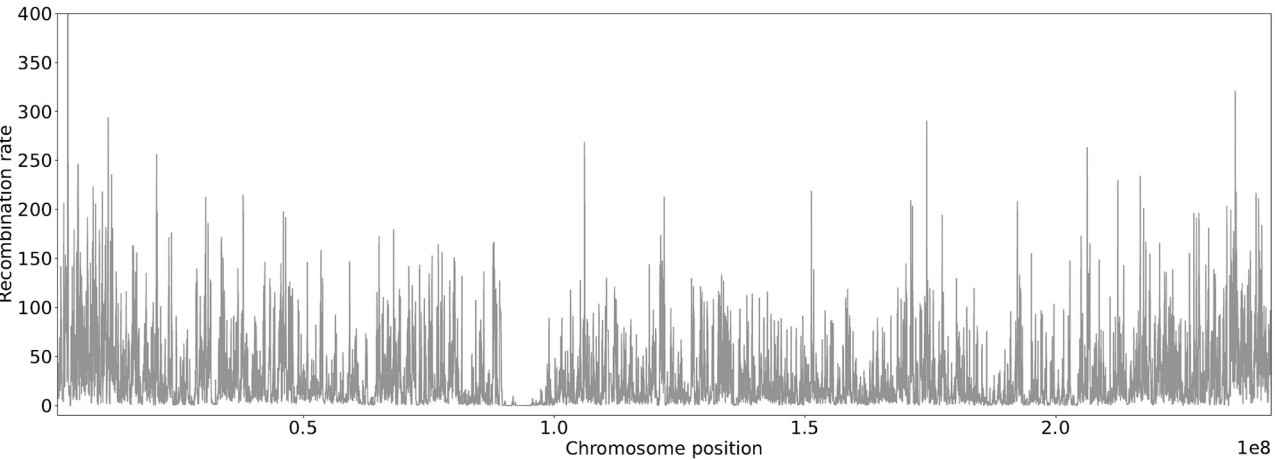

**Fig 4. Recombination map for human chr2.** Displayed recombination rates along the chromosome are based on a sliding window of 70kb. This also ensures better comparability with Fig 5. The mean recombination rate in the 70kb windows varies along the genome and is on average approximately $\rho = 31$.

## Discussion and conclusion

Different optimal estimators for the mutation rate or population size are known in scenarios of low and high recombination. Fu's and Watterson's estimator are linear unbiased estimators with minimal variance in the absence of recombination and in the limit of large recombination rates, respectively. However, simulations show that they do not adapt well to other frequencies of recombination. In the case of unknown recombination rates, it is therefore necessary for model-based estimators to estimate the recombination rate in a separate step and numerically compute a suitable estimator [12].

We were able to circumvent this step and created a simple neural network that has a low MSE regardless of the recombination rate. It is worth noting that the true recombination rate

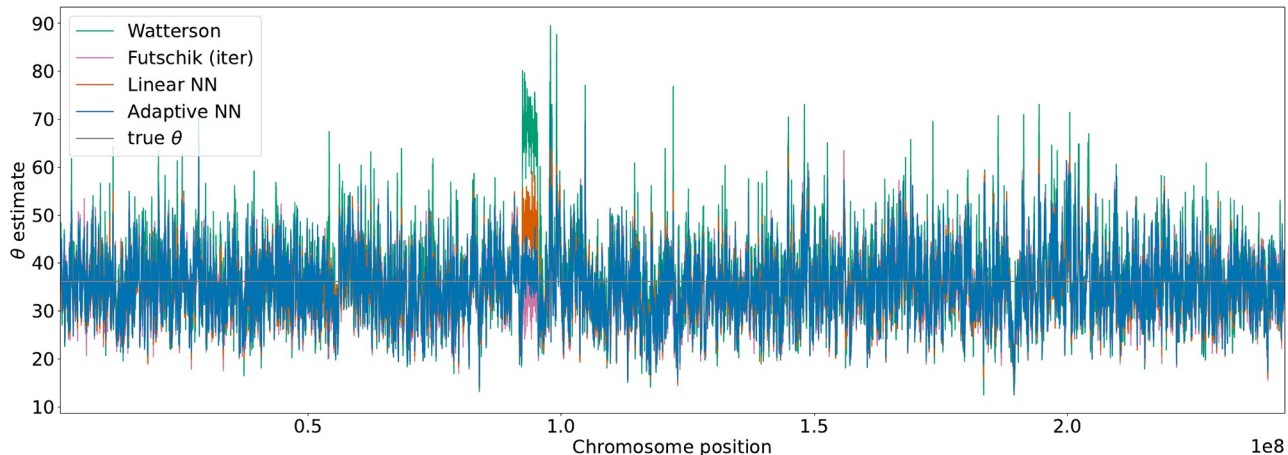

**Fig 5. Estimation of $\theta$ for human chr2.** The SFS for $\theta$ estimation was calculated based on a sliding window of size 70kb. Estimates based on Watterson, Futschik, the linear neural network and the adaptive neural network are displayed. The neural networks have been trained with variable recombination rates. The underlying true mutation rate $\theta = 36.12$ is shown as a grey line.

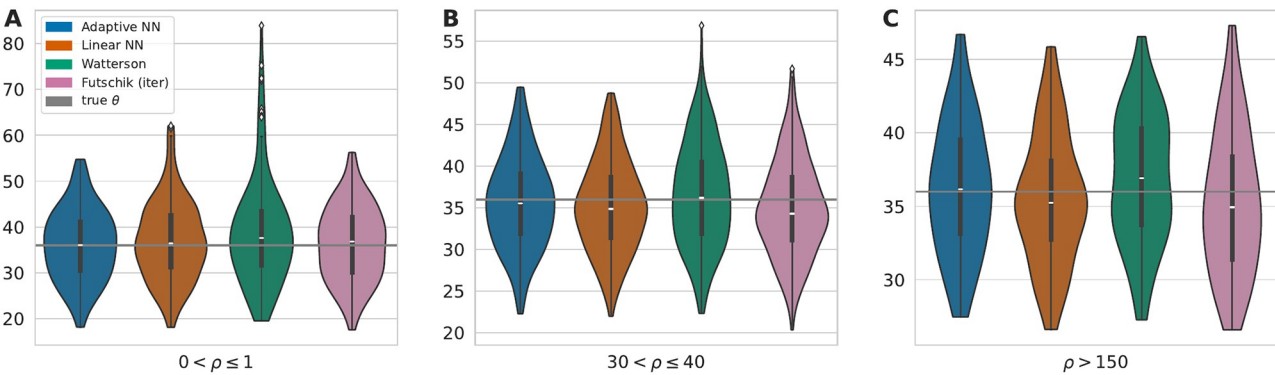

**Fig 6. Estimates of $\theta$ in regions of chr2 with low, intermediate, and high recombination.** The mean recombination rates of disjoint 70kb windows of chr2 were calculated and estimates for regions with low ($0 < \rho \le 1$) (A), medium ($30 < \rho \le 40$) (B) and large ($\rho > 150$) (C) recombination rates are shown in a Violin plot. The white marker in the box of the violins visualizes the median. The true mutation rate $\theta = 36.12$ is shown as a grey line.

was not used as an additional input here. Instead, the neural network adapts to the unknown recombination rate solely based on the observed SFS.

In contrast to the recombination rate, variability in the mutation rate is no issue in practice, although some of the considered estimators depend on the true but unknown mutation rate $\theta$. Iterative procedures for Fu's and Futschik's estimator are usually converging after 3–5 iterations and show almost the same performance as the optimal estimators (Fig H in S1 Text). In general, when training a neural network, the parameter range must be chosen carefully as they can learn the simulation range and consequently overestimate or underestimate outside this range. In our setting, this is mainly a problem if $\theta$ is smaller in the application than in the training data set since small $\theta$ values demand special attention to be learned. The linear and adaptive neural networks do not show dramatic overestimation or underestimation outside the training range, as Fig I in S1 Text demonstrates. In an application, if there is no idea about the realistic range of $\theta$, we would recommend first using the Watterson's estimator to get an approximate idea about the range of $\theta$ and retrain the neural network if necessary.

We observe that the neural networks approximate model-based estimators in situations where the model-based estimators are close to the optimal estimator (no or high recombination) and outperform them in scenarios that are hard to treat theoretically (moderate recombination). This remains true when applied in a more realistic setting with variable recombination along the human chr2. Even compared to more sophisticated ML methods, i.e. Flagel's CNN and ABC, the adaptive neural network remains the preferred choice for all recombination scenarios considered. In particular, the advantages of an adaptive method or training procedure become apparent, since otherwise, for example, small $\theta$ values are not given enough attention. This again highlights the capabilities of the adaptive neural network despite its architectural simplicity and the potential of using model-based results to adjust the training process. There is a multitude of other model-based estimators for the mutation rate, population size and related quantities, e.g. for the expected heterozygosity [34], that could enable machine learning methods in population genetics to adjust their performance.

Clearly, neural networks are computationally more demanding than model-based approaches. For example, it is necessary to train the network separately for each sample size. However, training is fast, even for the adaptive neural network. For $n = 40$ and training data with $2 \cdot 10^5$ simulated SFS, training is completed after approximately 20–50 iterations, which takes about 10 minutes on a laptop with an AMD Ryzen 7 octacore and 32 GB RAM. If these pitfalls are considered, even minimal artificial neural networks are attractive alternatives to

model-based estimators of the scaled mutation rate since they can perform uniformly well for different recombination scenarios.

It is worth noting that the observed outperformance of the neural networks was obtained solely based on the site frequency spectrum, but genetic data usually contains more information about the distribution of mutations among samples and their location along the genome. Machine learning methods based on the raw genetic data will require more complex network architectures and could help to automatically extract this additional information. Recent advances in the development of machine learning methods for population genetics [22, 23, 35–37], indicate that expanding the toolbox of neural network-based inference tools to more complex estimation tasks and larger data sets is a promising approach.

## Supporting information

**S1 Text. Supporting Information regarding the feature importance of neural networks, neural networks training procedure, application with a block recombination map, and further supporting figures.**
(PDF)

## Acknowledgments

We thank Philipp Harms for helpful discussions, Axel Fehrenbach for thoughtful input, and Libera Lo Presti for careful proof-reading of the manuscript.

## Author Contributions

**Conceptualization:** Klara Elisabeth Burger, Peter Pfaffelhuber, Franz Baumdicker.

**Formal analysis:** Klara Elisabeth Burger.

**Funding acquisition:** Franz Baumdicker.

**Methodology:** Klara Elisabeth Burger, Peter Pfaffelhuber, Franz Baumdicker.

**Software:** Klara Elisabeth Burger.

**Supervision:** Franz Baumdicker.

**Writing – original draft:** Klara Elisabeth Burger.

**Writing – review & editing:** Klara Elisabeth Burger, Peter Pfaffelhuber, Franz Baumdicker.

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
