## [Decision Letter · Decision Letter 0]

10 Jan 2022

Dear Dr. Baumdicker,

Thank you very much for submitting your manuscript "Neural Networks for self-adjusting Mutation Rate Estimation when the Recombination Rate is unknown" for consideration at PLOS Computational Biology.

As with all papers reviewed by the journal, your manuscript was reviewed by members of the editorial board and by several independent reviewers. In light of the reviews (below this email), we would like to invite the resubmission of a significantly-revised version that takes into account the reviewers' comments.

As you can see, the reviewers are very positive, albeit with some comments that should be addressed. We are particularly interested in seeing a biological application or case study (mentioned by both reviewers #1 & #3).

We cannot make any decision about publication until we have seen the revised manuscript and your response to the reviewers' comments. Your revised manuscript is also likely to be sent to reviewers for further evaluation.

Sincerely,

Luis Pedro Coelho

Associate Editor

PLOS Computational Biology

Thomas Leitner

Deputy Editor

PLOS Computational Biology

As you can see, the reviewers are very positive, albeit with some comments that should be addressed. We are particularly interested in seeing a biological application or case study (mentioned by both reviewers #1 & #3).

Reviewer's Responses to Questions

**Comments to the Authors:**

Reviewer #1: In this study, authors propose to estimate the population mutation rate ($\\theta = 4 N_e \\mu$) from the site frequency spectrum using neural networks. They compare the performance of their estimation against known estimators at several experimental conditions. The idea is of interest and it aligns with several ongoing efforts to use neural networks to estimate population genetic parameters.

I have some comments.

How would other deep learning algorithms that have been proposed recently compare against the one proposed in this study to infer the mutation rate? Also, since training is done by simulations, how would ABC perform in this case? It would be of interest to assess how much a simpler approach proposed here (MLP with 0 or 1 hidden layers from the SFS) compare against more sophisticated methods (e.g. CNN or RNN or GANs from either summary statistics or raw genomic data). In particular, how does it compare against Adrion et al. 2020 MBE? In general, it seems that more complex architectures have been proposed in popgen to infer more challenging parameters (e.g. full demographic histories rather than one estimate of theta). Therefore, it should be more evident the contribution to the field that this paper brings.

To clarify the scope of this study, it would be useful to deploy the train network to real data to infer the mutation rate on species/populations of interest to gain some biological insights.

I could not find details on the division of simulations between training, validation and testing data sets. Is the performance shown coming from the testing set? Also, can you show that you don’t have overfitting? How did you do hyperparameters tuning?

Regarding simulations, what is the unit of measure for the recombination rate? Is it scaled by N_e? What is the length of the region simulated? Would it be more realistic to simulate variable recombination rates that mimic known recombination maps (e.g. in humans).

The introduction and literature review are too succinct in my opinion. For instance, I was expecting more information of the biological context of the study: What is the mutation rate in various domains in life? How is regulated? How does it vary along the genome or between populations? What is the importance of inferring this value? Additionally, there are more papers using deep learning in population genetics and authors should also briefly survey the different architectures and input data used by these previously published studies.

Figure 2 is a cartoon of a generic MLP and it seems to me that it doesn’t provide any specific information. Likewise, Figure 1 is an illustration for calculating the SFS from a coalescent tree and it appears rather simplistic. I also wonder whether Figure 3 should be better presented as a formal algorithm, given the audience of this journal.

From Figure 4, it is clear that neural nets provide better estimates at moderate recombination rate, but what is the scale of this improvement? In other words, and for instance, how much biased downstream analyses (e.g. N_e) would be when using classic estimators instead of neural nets? This should give a better idea of the impact of the deep learning approach.

It should be made clearer in the methods what is novel and what is already know. From my understanding, everything from line 58 to 94 are derivations already known.

Additional points

It would be of interest to comment on the inference of other estimators of the population mutation rate based on gene diversity (rather than S) and their various extensions (as in https://academic.oup.com/mbe/article/26/3/501/976423), and how neural nets can tackle this problem.

Can you infer the optimal estimator for the trained neural nets? In other words, from the learned weights can you attempt any interpretation of what are the most important input units (the SFS) and their combination for the parameter to predict?

In the abstract, “For intermediate recombination rates, the calculation of optimal estimators is more involved”, do you mean “is more convoluted” or “challenging”?

The very first sentence is too long and convoluted for being the beginning of a paper.

Line 49: I’d say that there isn’t a limited theoretical insight in population genetics.

Line 113: n is number of chromosomal copies or diploids?

Is there any scope to jointly infer mutation and recombination rates?

Is it possible to do an exhaustive architecture search to find the optimal set up for this study?

Reviewer #2: This paper by Burger et al. develops some fairly simple artificial neural networks as alternatives to model-based estimators of theta. This is a relevant topic because existing estimators do well in the case of free recombination or no recombination, but struggle in intermediate cases. The authors show that their neural network approach yields a more general-purpose estimator. I found this paper interesting and well-written, and have only a number of relatively minor comments that I would like to see addressed prior to publication:

-Line 138: "It is advisable to use at least 2n nodes." It would be more helpful if the authors could share any analyses that support this claim.

-Between Lines 150-151: Can the authors explain how t_k are chosen? (Referring the reader to Figure S4 might also help here somewhat, but this Figure is not currently referenced in the text.)

-Also, the condition that the authors are trying to satisfy when choosing the t_k includes the term a_j(t_{k-1}). When k=1, this gives a_j(t_0), but I don't think this is defined.

-Line 155: The authors state that training finishes only once the network performs "comparably or better than the model-based estimators and the linear NN on each of the six test subsets." I have two comments on this: 1) By comparable or better, the authors mean "exactly equal to or better", correct? 2) Is this always guaranteed to happen? I would imagine that, especially if one were to test a neural network with low capacity, the network may not always be able to achieve this. Did this ever happen when the authors were testing out different architectures? I think some readers would find this information very useful and interesting.

-Line 179: The authors kind of downplay the performance of the linear neural net here. Looking at the results, it seems that this simple network has done quite well! I would consider rephrasing the text here to more accurately describe these results.

-Figure 4 caption: "For each shown data point 10000 simulations with sample size n = 40..." How many data points are shown? This would help the reader determine how many total test simulations were generated for this figure.

-Lines 203-205: Here the authors state that the neural nets "dramatically over or underestimate the mutation rate" outside of the training range of this parameter. But the performance in S2 Fig does not look too bad to me. It seems like the neural nets perform about as well as the best model-based estimator for each case. So, while the error rates increase, it doesn't seem that there is some dramatic collapse once we get outside of the training range, unless I am missing something. Perhaps less pessimistic language would be appropriate here.

-In any case, it is clear that error rate does increase as we move further and further from the training range. What should one do about this if for some population we really have no idea what theta is? How would the authors deal with this in practice?

Line 213: What is a standard laptop? Please share the specs if you don't mind.

Reviewer #3: The authors consider the classical problem of estimating the scaled mutation parameter in a population genetic context. They consider feedforward neural networks and show that they are able to achieve a

performance comparable to previously proposed estimates both for high and low recombination rate. Indeed, the results seem to indicate that neural networks are able to adapt to an unknown recombination rate, whereas with classical estimators, the correct one has to be picked depending on recombination. Interestingly, the neural networks work well, even without a hidden layer.

The obtained results are interesting and contribute to an understanding of potential benefits of machine learning methods compared to mathematical theory based approaches.

On the other hand, in practical applications there is usually an idea about the underlying recombination rate. (And if not it can also be estimated.)

Therefore the actual gains achieved by machine learning in this context remain unclear. There seem to be some slight advantages of neural networks at intermediate recombination rates, but it would be nice to see a biological example, illustrating that there is actually an organism and a segment length for which the intermediate recombination (and mutation) rates are realistic. It would also be interesting to consider more complex scenarios and see whether neural networks provide larger advantages there (maybe with demography?).

In general, the manuscript is well written and structured.

Further comments:

* The authors claim in the abstract and at other places that the Fu’s estimate is optimal. This is not correct: Fu shows optimality (BLUE, best linear unbiased estimate) only for a variant of his estimate that requires knowledge of the true unknown mutation rate. But if the mutation parameter is known, there is no need to estimate it anymore. The variant where an estimate of theta is used instead is neither linear nor unbiased. And as far as I know, there is also no proven optimality result for this estimate. The corresponding claims should be rephrased accordingly.

* It is possible to apply estimates on DNA segments short enough so that recombination can be ignored. For constant recombination, these local estimates can be averaged. How does such a strategy work in the case of intermediate recombination rates.

**Have the authors made all data and (if applicable) computational code underlying the findings in their manuscript fully available?**

Reviewer #1: Yes

Reviewer #2: Yes

Reviewer #3: Yes

PLOS authors have the option to publish the peer review history of their article (what does this mean?). If published, this will include your full peer review and any attached files.

Reviewer #1: **Yes: **Matteo Fumagalli

Reviewer #2: No

Reviewer #3: No
---

## [Decision Letter · Decision Letter 1]

14 Jun 2022

Dear Dr. Baumdicker,

Thank you very much for submitting your manuscript "Neural Networks for self-adjusting Mutation Rate Estimation when the Recombination Rate is unknown" for consideration at PLOS Computational Biology. As with all papers reviewed by the journal, your manuscript was reviewed by members of the editorial board and by several independent reviewers. The reviewers appreciated the attention to an important topic. Based on the reviews, we are likely to accept this manuscript for publication, providing that you modify the manuscript according to the review recommendations.

The major scientific concerns have been addressed, but we ask that the authors clarify the minor points raised by the reviewers.

Sincerely,

Luis Pedro Coelho

Associate Editor

PLOS Computational Biology

Thomas Leitner

Deputy Editor

PLOS Computational Biology

[LINK]

The major scientific concerns have been addressed, but we ask that the authors clarify the minor points raised by the reviewers.

Reviewer's Responses to Questions

**Comments to the Authors:**

Reviewer #1: Authors now compare their method with competing strategies. Whilst they do not deploy the network to real data, they use realistic recombination maps for their simulations. The introduction is now more comprehensive. Details or explanations that were missing are now present. Minor things have been fixed as suggested. The manuscript has significantly improved.

As a minor comment, I’d suggest the Authors to consider colourblind-friendly colour-palette (e.g. for Fig 6).

Also the title “comparison to more sophisticated ML Methods” can be changed to simply “comparison with alternative/competing/existing ML methods”. Is ABC an ML method? In case, the “ml” in the title can be dropped.

In Fig 5 different lines are hard to see.

Reviewer #2: The authors have done a thorough job addressing my (mostly minor) comments, and the paper is suitable for publication as is.

Reviewer #3: This new version of the manuscript is quite pleasant to read. The revision has addressed my concerns.

I have only two minor points.

l. 113-114: What do you mean by not uniform? The actual estimator with estimated theta is neither linear nor unbiased, and there is no proof that it is unbiased.

l. 121-123: There are actually multiple variants of the estimate proposed by Futschik & Gach. One of them is a modification of Watterson's estimate and does not depend on theta. There are also iterative versions, both of Fu's estimate and the Watterson estimate. This should be mentioned and it should be explained more clearly which of the variants has been used. It would be nice to show also the performance of the non-iterative modification of Watterson's estimate, maybe as supplementary material.

**Have the authors made all data and (if applicable) computational code underlying the findings in their manuscript fully available?**

Reviewer #1: Yes

Reviewer #2: None

Reviewer #3: Yes

PLOS authors have the option to publish the peer review history of their article (what does this mean?). If published, this will include your full peer review and any attached files.

Reviewer #1: **Yes: **Matteo Fumagalli

Reviewer #2: No

Reviewer #3: No

Figure Files:

Data Requirements:

Reproducibility:

References:

---

## [Editor Report · Decision Letter 2]

18 Jul 2022

Dear Dr. Baumdicker,

We are pleased to inform you that your manuscript 'Neural Networks for self-adjusting Mutation Rate Estimation when the Recombination Rate is unknown' has been provisionally accepted for publication in PLOS Computational Biology.

Best regards,

Luis Pedro Coelho

Associate Editor

PLOS Computational Biology

Thomas Leitner

Deputy Editor

PLOS Computational Biology

---

## [Editor Report · Acceptance letter]

31 Jul 2022

PCOMPBIOL-D-21-01626R2 

Neural Networks for self-adjusting Mutation Rate Estimation when the Recombination Rate is unknown

Dear Dr Baumdicker,

I am pleased to inform you that your manuscript has been formally accepted for publication in PLOS Computational Biology. Your manuscript is now with our production department and you will be notified of the publication date in due course.

With kind regards,

Zsofia Freund
